# Is Random Attention Sufficient for Sequence Modeling?

## Abstract

The transformer architecture is central to the success of modern Large Language Models (LLMs), in part due to its surprising ability to perform a wide range of tasks – including mathematical reasoning, memorization, and retrieval – using only gradient-based learning on next-token prediction. While the core component of a transformer is the self-attention mechanism, we question how much, and which aspects, of the performance gains can be attributed to it. To this end, we compare standard transformers to variants in which either the attention weights or the MLP layers are frozen at initialization. Surprisingly, we find that attention with *frozen* key and query weights is not only able to form induction heads, but can also perform competitively on language modeling. We formalize this by proving a new expressivity result for transformer models with frozen attention weights. To further isolate the contribution of attention, we design MixiT – the Mixing Transformer – an architecture variant with entirely random attention scores, with provably stable signal propagation that overcomes prior depth-wise scaling challenges in random transformers. We use the successes and failures of our spectrum of models to pinpoint the role each main transformer component plays. Our results suggest that the transformer architecture has a built-in inductive bias towards in-context reasoning, as it can form specialized circuits even without learnable attention weights.

## 1 Introduction

Transformers (Vaswani et al., 2017) have rapidly become the workhorse architecture in modern machinelearning systems, powering stateoftheart models in language, vision, and scientific domains (Dosovitskiy et al., 2020; Team et al., 2023; Guo et al., 2025). Their success is typically attributed to the selfattention mechanism, which allows every token to aggregate information from the entire sequence and has been linked to emergent abilities such as longrange retrieval, algorithmic reasoning, and incontext learning. Yet we lack a precise answer to a fundamental question: which degrees of freedom inside the transformer are truly necessary for these behaviors, and which can be simplified away without harming performance?

Prior work has probed the internals of trained transformers. Studies of attention maps consistently report the emergence of induction heads that copy information forward and enable incontext retrieval (Olsson et al., 2022). Another line of work has focused on whether and how transformers' capabilities can emerge (Allen-Zhu and Li, 2024; Ramesh et al., 2023; Jain et al., 2023) by studying synthetic datasets, with Zhong and Andreas (2024) showing that models with frozen weights but trainable embedding and unembedding layers can still solve certain algorithmic tasks. These results hint that different parts of the architecture are responsible for modeling different tasks.

A standard transformer block, however, contains several interacting components: an attention block, with learnable, input-dependent queries and keys, and MLP blocks composed of fully connected layers. To exemplify the complexity of the interaction, notice that the attention weights can change their value both through changes in the queries/keys parameters, as well as as through the hidden representations that feed those projections. Hence, as all the components are trained together, it is challenging to discern the contribution of each individual one towards solving a given task. In this work, we study the role of each transformer's component by freezing different parts of the architecture to their values at initialization. In particular, we consider the following architectural variants:

- *FrozenQK*, preserves the conventional attention structure but freezes the query and key weight matrices.

- *Frozen-MLP*, where the weight matrices of the MLP block are frozen at random initialization.

- *MixiT* (Mixing Transformer), where the attention scores matrix are fixed at a principled random initialization, designed to ensure stable signal propagation during the forward pass. In particular, the attention matrices are entirely input-independent.

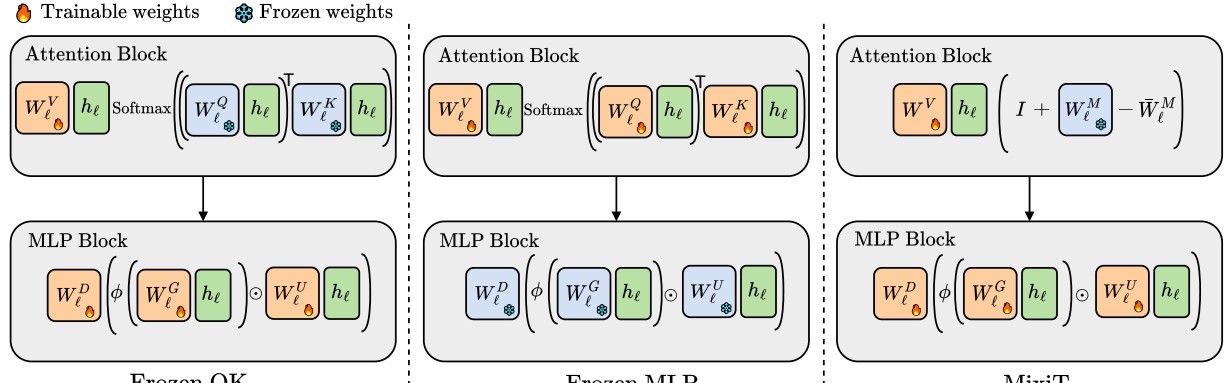

Figure 1: Variants of the transformer architecture in our study. The successes and failures in different tasks for this spectrum of models allow us to pinpoint the functionality of each main model component with respect to different tasks.

This spectrum of models allows for rich analyses on the attention architecture, such as separating the role of learned token-dependent weights, versus the mere presence of a mixing operation. We compare these architectual variants across several categories of tasks: mathematical reasoning, sentiment classification, memorization, in-context reasoning such as retrieval and $k$-hop, and language modeling. This allows us to inspect whether and how the different parts of the architecture are able to solve tasks that relate to a range of basic reasoning and memorization skills. Our main contributions are:

- We find that, surprisingly, Frozen-QK, which has random attention weights, can perform competitively with the standard transformer on language modeling tasks. Indeed, Frozen-QK develops the ability to form specialized circuits such as induction heads during training. This suggests that learnable attention weights are *not* required to form specialized circuits.

- We analyze Frozen-QK's expressiveness, and show that it can indeed learn a wide class of sequence-level functions in Theorem 5.1.

- We identify an explanation of the subspace selection hypothesis for random transformers (Zhong and Andreas, 2024), namely, representation collapse, and link it to the phenomenon that random models with standard initializations suffer degenerating performance with respect to depth.

- To resolve this obstruction towards scaling up model depth, we introduce a principled architecture – MixiT, as part of our spectrum of models, and prove its training stability, by analyzing its covariance SDE, in Theorem 2.1.

- We find that, in all but induction heads tasks and language modeling, MixiT achieves performance comparable to the fully trained transformer and Frozen-QK. These results suggest that for a wide range of tasks, "attention mixing is all you need" regardless of its specific form, in the sense that learned querykey interactions are not required.

Our results indicate that the transformer architecture contains built-in inductive bias towards algorithmic abilities, namely the ability to form specialized circuits, as evidenced by the presence of induction heads even without learnable attention weights. Furthermore, MixiT can serve as a *litmus test* for whether a task requires in-context reasoning.

## 2 MODELS SPECIFICATION

We consider a decoder-only Transformer based on the widely used Llama architecture (Touvron et al., 2023). Given an input sequence $x \in \mathbb{R}^{V \times m}$, where $V$ is the vocabulary size and $m$ is the sequence length, embedded with a linear map to the hidden states $h_0 = W_{emb} x$ where $W_{emb} \in \mathbb{R}^{V \times n}$, where $n$ is the width of the model. At its core, the transformer uses $L$ stacked modules alternating the causal multi-head self-attention layers and MLP layers. Each self attention head is defined as:

$$\text{Attn}(h_\ell) = W_\ell^v h_\ell \, \text{Softmax}\left(\frac{1}{\sqrt{n_h}} Q_\ell^T K_\ell\right), \quad Q_\ell = W_\ell^Q h_\ell, \quad K_\ell = W_\ell^K h_\ell, \tag{2.1}$$

where $W_\ell^Q, W_\ell^K, W_\ell^v \in \mathbb{R}^{n_h \times n}$ are the queries' and keys' weights, and the outputs across multiple heads are concatenated. The gated MLP layer is defined as:

$$\text{MLP}(h_\ell) = W_\ell^D \left( \phi(W_\ell^G h_\ell) \odot W_\ell^U h_\ell \right) . \tag{2.2}$$

where $W_\ell^G, W_\ell^U \in \mathbb{R}^{n_m \times n}$ and $W_\ell^D \in \mathbb{R}^{n \times n_m}$. $n_h$ and $n_m$ are the dimensions of the queries/keys for each head and MLP projections. We also use skip connections in both blocks with a pre-normalization scheme (Xiong et al., 2020) and causal masking. We apply rotary embeddings (Su et al., 2024) to the queries and keys of each layer. As in common practice, we use $n_m = 4n$ and $Hn_h = n$.

**Models with Frozen Weights.** In the Frozen-QK model, we set $W_\ell^Q, W_\ell^K$ to their values at initialization, and for the Frozen-MLP models we freeze $W_\ell^D, W_\ell^G, W_\ell^U$ for all layers.

**MixiT – Random Static Attention.** We also design a model where the attention map itself is frozen and, to achieve that, input-independent. In the simplest case, this be obtained by having a random matrix $M_\ell \in \mathbb{R}^{m \times m}$ entries with $\mathcal{N}(0, 1/m)$ entries, where the factor of $1/m$ acts as a variance-preserving normalizer. To ensure a stable forward pass in terms of depth and width scaling, we follow the principles of attention shaping (Noci et al., 2023) and propose the following:

$$\text{Attn}(h_\ell) = W_\ell^v h_\ell \left( I + W_\ell^M - \bar{W}_\ell^M \right) , \quad W_{\ell,ij}^v \overset{iid}{\sim} \mathcal{N}(0, \tfrac{1}{n}), W_{\ell,ij}^M \overset{iid}{\sim} \mathcal{N}(0, \tfrac{1}{\sqrt{nm}}), \tag{2.3}$$

where $W_\ell^M$ is frozen at initialization and $\bar{W}_\ell^M$ contains the column-wise empirical average of $W_\ell^M$, to ensure that each row sums up to 1. In Appendix A, we show that this attention variant has a stable forward pass, in the sense that the kernel of the activations has a well-defined depth-and-width limit, converging to a stochastic differential equation (SDE) (Li et al., 2022; Noci et al., 2023). Specifically, we prove that the covariance of the hidden representations has a stable limit, and converges to the solution of this SDE. The stability of the covariance has been well-established to be necessary for stable training (Xiao et al., 2020; Schoenholz et al., 2016; Murray et al., 2022; Hayou et al., 2019; Yang and Schoenholz, 2017b; Poole et al., 2016). When we adopt this architecture, all weights other than the random attention matrix are trainable. Notably, the following convergence result implies the stability of the forward pass, in particular ruling out the numerical degeneracy such as rank collapse and vanishing gradients (Dong et al., 2021; Noci et al., 2022). The exact statement and the proof can be found in Appendix A.

**Theorem 2.1** (MixiT Covariance SDE (Abbreviated)). *Consider the MixiT recursion $h_{\ell+1} = Attn(h_\ell)$ defined by (2.3) at initialization. Then as the width $n$ and depth $d$ go to infinity with $\frac{d}{n} \to \bar{\tau} > 0$, the upper triangular entries of the covariance matrix $\Phi_\ell = \frac{1}{n} h_\ell^\top h_\ell$ flattened to a vector in $\mathbb{R}^{m(m+1)/2}$ converges to the solution of the following SDE*

$$d\Phi_\tau = \left[ \frac{1}{m} \text{Tr}(\Phi_\tau) - M(\Phi_\tau) \right] d\tau + [\Sigma^v(\Phi_\tau) + \Sigma^M(\Phi_\tau)]^{1/2} dB_\tau , \tag{2.4}$$

*where $M(\Phi) = \frac{1}{m^2} \sum_{\alpha\beta=1}^m \Phi^{\alpha\beta}$ is the average over all entries, $B_\tau$ is a standard Brownian motion in $\mathbb{R}^{m(m+1)/2}$, and $\Sigma^v(\Phi_\tau)$ and $\Sigma^M(\Phi_\tau)$ are diffusion coefficients, defined exactly in Section A.*

Indeed, this provably stable forward pass implementation helps MixiT scale well with respect to depth, and helps explain why some random-weight transformer variants collapse, discussed in detail in Section 5. Note that we regard MixiT both as an improved, more principled random transformer, and as a tool for understanding the standard transformer, not as a state-of-the-art model architecture.

**Positional embedding.** As the random attention matrix $I + \frac{1}{\sqrt{mn}} W_\ell^M - \bar{W}_\ell^M$ in MixiT does not depend on the input, the rotary positional embedding (Su et al., 2024) widely used in transformer models cannot be applied to MixiT, as rotary embeddings are added to learned key and query embeddings, which are not used in MixiT. Hence, we implement a learnable positional embedding for each token position in the sequence, and add it to the corresponding token embedding in the first layer.

# 3 EXPERIMENTS

## 3.1 TASKS STUDIED

We benchmark our model variants on 8 tasks spanning a variety of categories, including mathematical reasoning, memorization, in-context reasoning, and language modeling. The mathematical reasoning and memorization tasks are based on tasks used in (Zhong and Andreas, 2024), with increased difficulty on some tasks to better reflect differences between architectures.

**Decimal Addition.** For the decimal addition task, the model learns to add two integers with the same number of digits. We randomly sample 50,000 pairs of ten-digit numbers, and train the model to predict their sum. The test set consists of 4,000 such sequences not in the training set. An example is $1234567890 + 2345678901 \rightarrow 3580246791$.

**Needle in a Haystack (Retrieval).** Each training instance encodes a small randomly generated sequence of pairs followed by a single query, and the model is required to emit the value associated with that query one step later. We uniformly sample a sequence length $m \sim \mathcal{U}\{1, \ldots, m_{\max}\}$, where $m_{\max}$ is the maximum sequence length. We then sample $m$ *keys* $\{k_i\}_{i=1}^m$ iid from the set $\{\frac{V}{2}, \ldots, V\}$ and $m$ *values* $\{v_i\}_{i=1}^m$ from $\{1, \ldots, \frac{V}{2} - 1\}$, where $V = 256$ is the vocabulary size. The resulting keys are interleaved with their values to form the prefix $[(k_1, v_1), (k_2, v_2), \ldots, (k_m, v_m)]$. A query key $k_q$ with $q \in [m]$ is chosen uniformly at random from the keys, and appended to the sequence. The goal is to predict the value corresponding to the query token. This task isolates retrieval ability, and probes associative recall. We sample 40,000 sequences for training, and 4,000 for testing. Transformer-based models typically solve this task by forming induction heads (Olsson et al., 2022).

**$k$-hop Induction Heads.** Following (Sanford et al., 2024), the $k$-hop induction heads task, or hop$_k$, recursively completes bigrams auto-regressively, by repeatedly predicting the token that followed the last occurrence of the currently-considered token. As an example: given the input $X = adcada$, the 2-hop induction heads prediction is $c$. (Sanford et al., 2024) showed that hop$_k$ is solvable by a $O(\log k)$-depth transformer. Hence to achieve fair comparison across architectures, we fix the model depth at 5 layers, and search over other hyperparameters.

**Modular Addition.** This task evaluates the model's ability to perform addition, modulo a given prime $p$. In our case we sample 40,000 pairs $(a, b)$ of integers, each within the range $[1, p]$, for $p = 599$, and train the model to predict $a + b \bmod p$.

**Parentheses Balancing (Dyck-1).** The parentheses balancing task learns to predict whether a given parentheses sequence is balanced. Within any prefix in the sequence, the number of closing parentheses is less than or equal to the number of opening parentheses. Hence this task is solvable by a linear time algorithm. We randomly sample 100,000 parentheses sequences of length 40 for training, while the test set consists of 4,000 such sequences not in the training set. An example is "(()" $\rightarrow$ False.

**Memorization.** We follow the procedure in Zhong and Andreas (2024): we sample $512^2$ key-value pairs, where each key is independently sampled from its value, which is an integer in $[512]$. Because the key-value mapping is random, any success reflects the models ability to memorise arbitrary associations. We measure success with the number of *bits per parameter* that the model can store, defined as `#total_bits` $\times$ `model_acc/total_trainable_params`, where `model_acc` is the training accuracy on the task. Notice that for this problem, storing one pattern requires $\log_2 512 = 9$ bits, thus `#total_bits` $= 9 \cdot 512^2$.

**Sentiment Classification.** We use the Yelp polarity reviews dataset (Zhang et al., 2015) to test each model variation's ability to predict a review's sentiment, i.e. whether a review is positive or negative.

**Language Modeling.** To test the model's ability to model natural language, we train the model to perform next-token-prediction. We use two datasets, Wikitext-103 (Merity et al., 2016) and Fineweb-edu (Penedo et al., 2024). Wikitext-103 consists of 1,801,350 cleaned-up Wikipedia articles, with a test set of size 4358. And Fineweb-edu consists of top-quality entries collected from web-crawled data, focusing on educational text. We randomly sample 1,048,576 entries for training, and 4096 to test.

## 3.2 MODEL TRAINING

For each task, we perform a grid search over a range of hyperparameters to train all model variations. The optimal hyperparameters are determined using a grid search specific for each task, which are detailed in the Appendix C. All model variants are trained on one to eight H100 GPUs, depending on task complexity.

## 4 RESULTS

**Frozen-QK can solve induction heads tasks such as retrieval and $k$-hop.** We test our spectrum of models on tasks that require forming induction heads (Elhage et al., 2021): needle-in-a-haystack retrieval (Olsson et al., 2022) and $k$-hop induction heads ($k$-hop) (Sanford et al., 2024). As shown in Table 1, Frozen-QK is able to solve these tasks on

par with the standard fully trained transformer model, indicating that induction heads can form even with frozen query and key weights. On the other hand, MixiT is unable to solve these tasks, due to its inability to form specialized circuits as the attention scores are frozen at random initialization.

| Task \ Model | Standard | Frozen-MLP | Frozen-QK | MixiT |
|---|---|---|---|---|
| $k$-hop Induction Heads ↑ | 99.99% | 99.89% | 96.73% | 48.58% |
| Retrieval ↑ | 100% | 100% | 97.01% | 11.24% |

Table 1: Accuracies for tasks that require forming induction heads. Frozen-QK is able to solve the tasks on par with the standard fully trained transformer model, reinforcing the observation that induction heads can form even with frozen query and key projectors, e.g. as evident in Figure 3. Furthermore, as the attention module plays the key role in forming induction heads (Elhage et al., 2021; Crosbie and Shutova, 2025), Frozen-MLP, with its trainable attention modules, is able to perform to almost perfect accuracy. On the other hand, MixiT is unable to solve these tasks, due to its inability to form specialized circuits as the attention scores are frozen at random initialization. The retrieval results are for sequences of maximum number of key-value pairs $m_{\max} = 30$.

More detailed results on the retrieval task are shown in Figure 2, where we test the performance of each architecture at varying task complexities, controlled by the maximum number of key-value pairs $m_{\max}$ in the sequence. As shown, Frozen-QK, Frozen-MLP, and the standard transformer all perform notably better than MixiT as the retrieval complexity increases; and Frozen-QK eventually deteriorates in quality faster than Frozen-MLP and the standard transformer, underscoring the key role attention plays in forming induction heads.

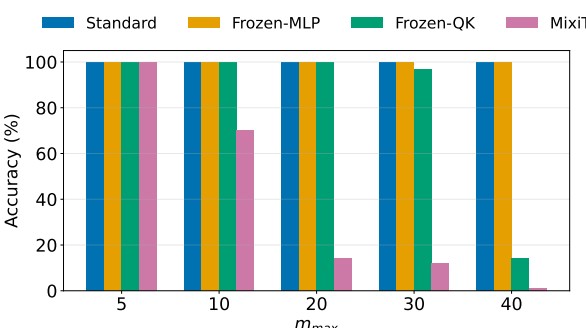

Figure 2: Retrieval accuracy as a function of the number of pairs in the sequence $m_{\max}$, which encodes task complexity. While MixiT has rapidly deteriorating performance with respect to task complexity, Frozen-QK, with its ability to form induction heads, reaches its performance ceiling much more slowly.

**Frozen-QK can perform competitively on language modeling.** Surprisingly, Frozen-QK comes close to the standard transformer in terms of perplexity, as shown in Table 2. This indicates that trainable attention weights are *not* always required for successful language modeling. Indeed, as Figure 3 shows, specialized circuits such as induction heads can form even in Frozen-QK. Less surprisingly, MixiT lags behind the standard Transformer, supporting the hypothesis that input-dependent learned attention patterns, such as induction heads, are necessary for language modeling, corroborating earlier works (Olsson et al., 2022; Crosbie and Shutova, 2025).

**Random static attention can perform certain algorithmic tasks.** Table 3 shows the models' performance on algorithmic and sentiment classification tasks. Both Frozen-QK and MixiT are able solve such tasks. They are competitive with, and can even outperform, the standard fully trained transformer. These results highlight that input-dependent attention is not required for solving such algorithmic tasks. In addition, comparing these positive results with MixiT's failure on retrieval and $k$-hop, where specialized circuits are required, imply that MixiT's performance on a task can serve as a litmus test on the existence of a solution without in-context reasoning.

**MLPs are crucial, and collaborate with attention, on memorization.** Table 4 shows the accuracy and the storage capacity via *bits per parameter*. We find that the standard transformer stores 2.98 bits per parameter, which is slightly higher than in previous works (Allen-Zhu and Li, 2024; Zhong and Andreas, 2024). Most of the drop occurs in the

| Model \ Task | Wikitext ↓ | Fineweb-edu ↓ |
|---|---|---|
| Standard | 2.78 | 3.51 |
| Frozen-QK | 3.07 | 3.64 |
| MixiT | 3.73 | 4.27 |

Table 2: Performance on language modeling tasks, in terms of log perplexity. Frozen-QK comes surprisingly close in performance to the standard Transformer, despite having random static query and attention weights. MixiT has notably worse performance, supporting the hypothesis that input-dependent learned attention patterns, such as induction heads, are necessary for good language modeling.

| Model \ Task | Decimal Addition↑ | Dyck-1↑ | Modular addition↑ | Memorization↑ | Yelp↑ |
|---|---|---|---|---|---|
| Standard | 98.58% | 95.80% | 100% | 100% | 90.55% |
| Frozen-QK | 100% | 97.38% | 100% | 100% | 90.86% |
| MixiT | 100% | 96.17% | 100% | 100% | 92.56% |

Table 3: MixiT performance on algorithmic and sentiment classification tasks. As shown, both Frozen-QK and MixiT are able solve such tasks. They are competitive with, and can even be superior to, the standard transformer.

Frozen-QK model, with 1.13 bits per parameter, while Frozen-MLP and MixiT have similar storage capabilities at 2.25 and 2.18, respectively. Note that these results are yielded in a setting where the accuracies are not saturated at 100%, to give an accurate representation of bits per parameter. Hence they do not contradict the results in Table 3.

| Model | Memorization Accuracy ↑ | Bits Per Parameter ↑ | Trainable Parameters |
|---|---|---|---|
| Standard | 100% | 2.98 | 790400 |
| Frozen-MLP | 19% | 1.13 | 394880 |
| Frozen-QK | 69% | 2.25 | 724352 |
| MixiT | 67% | 2.18 | 724736 |

Table 4: Standard transformers outperform all the alternatives in terms of memorization capability, even when adjusted for parameter count, which suggests that MLPs and attention *collaborate* to remember knowledge. This provides further evidence for recent findings such as knowledge circuits (Yao et al., 2024) and query localization (Chen et al., 2025). Freezing the MLPs causes the most performance drop, indicating that they are the biggest factor when it comes to memorization. Notice that MixiT has slightly more parameters than Frozen-QK because of additional trainable positional embeddings.

These results suggest that (1) the MLPs are largely responsible for memorization, however (2) learnable attention non-trivially assists MLPs in knowledge storage, as the bits-per-parameter drops from 2.98 for the standard transformer to 2.25 for Frozen-QK. Attention's non-negligible contribution provides further evidence for recent findings such as knowledge circuits (Yao et al., 2024) and query localization (Chen et al., 2025), in that MLPs and attention *collaborate* to remember knowledge. In particular, the disproportionately large increase in the learned bits per parameter from Frozen-QK to the fully trained transformer, from 2.25 to 2.98, suggests that the gain in accuracy is more than what can be accounted for by a mere increase in learnable parameters.

## 5 DISCUSSION

**Circuit learning and task separation.** As the attention matrix in MixiT is static and input-independent, MixiT cannot adapt to each input and form specific circuits such as induction heads (Olsson et al., 2022). Induction head circuits look for recent occurrences of the current token, and attends to the tokens that follow with increased probability. This allows the standard transformer to easily adapt to the input context in language modeling. Hence, it is no surprise that MixiT lags behind standard transformer for language modeling. It is perhaps more surprising that the perplexity comes close to that of the standard transformer.

The near-perfect performance on certain algorithmic tasks in Table 3 suggests that induction heads and other specialized input-dependent circuits are not required on these tasks. Hence the MixiT performance on a given task can serve as a litmus test for whether in-context reasoning is required for that task. For instance, MixiT is able to perform well on the Yelp reviews dataset, despite the complexity of language used in reviews. This indicates that sentiment can be largely judged by the collective token embeddings, as opposed to next-token prediction in language modeling tasks, which requires retrieving specific details from the context, a task induction heads are apt at.

**Learnable attention is not required to form induction heads.** Interestingly, as shown in Figure 3 and demonstrated by its performance shown in Table 1 and Table 2, the Frozen-QK model can solve the retrieval task by forming induction heads.

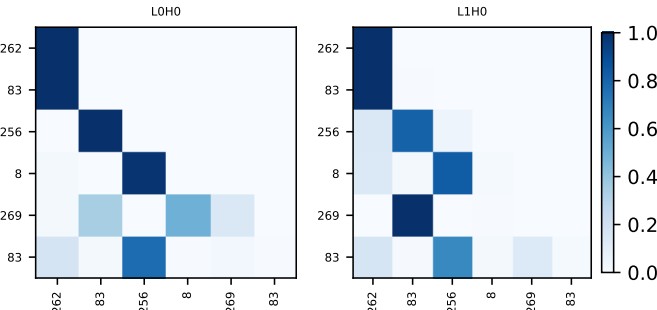

Figure 3: The Frozen-QK model can solve the retrieval task by forming an induction head. In the first head, each token attends to the previous one; in particular, the query token 83 is attended by 256. In the head of the second layer, the correct token 256 is retrieved.

These results naturally raise the question: how expressive is Frozen-QK? To answer this, we prove the following result showing Frozen-QK can approximate a wide class of functions:

**Theorem 5.1** (Universal Approximation of Frozen-QK). *Every continuous causal function with compact support can be approximated arbitrarily well by one layer of multihead attention and MLP, where query and key weight matrices are frozen at random initialization.*

The proof can be found in Appendix B. In summary, the proof leverages the fact that standard transformers are universal approximators of such functions (Yun et al., 2020), and lifts this universal approximation to random feature transformers. Note that in practice, there are also the value weights and the MLP layers in the attention module, so we do expect the representation power to be even stronger, which is evidenced by empirical results.

Furthermore, the proof sheds light on why MixiT cannot be a universal approximator: as $Q_k$ and $K_k$ are input-*independent* in MixiT, the random feature $g_i = x \, \text{Softmax}(Q_k^\top K_k)$ in MixiT is *linear*. Hence non-linear functions in $x$, such as the induction head function, cannot be approximated.

**Role of MLPs in knowledge storage.** Previous works have highlighted the importance of MLPs in storing knowledge (Dai et al., 2022; Geva et al., 2023; 2022; 2021; Yu and Ananiadou, 2024; Chughtai et al., 2024; Meng et al., 2022; Merullo et al., 2024). They posit that specific facts from the training data are stored in specific knowledge neurons. These works support our findings, in that MLPs are crucial in knowledge memorization. However, our work does not prescribe knowledge localization, i.e. we don't attribute memorizing specific facts to specific neurons.

Our work adds characterization on knowledge memorization in more recent works on knowledge circuits (Yao et al., 2024) and query localization (Chen et al., 2025), where MLPs and attention are found to *collaborate* on knowledge memorization, e.g. where attention selects the appropriate knowledge neurons depending on the query. Our work shows that even with static random attention weights, such as in Frozen-QK, attention and MLPs can still collaborate effectively, as evident in language modeling perplexities similar to that of the standard transformer, and the formation of specific input-dependent circuits. However, our results show that the role of MLPs in memorization outweighs that of attention, as evident by the fact that Frozen-MLP achieves much worse accuracy than Frozen-QK or MixiT (4).

**Relation to random transformers.** Zhong and Andreas (2024) studies the random transformer, wherein they train only the embedding and unembedding layers, and leave the intermediate layers fixed at random initialization. The

random transformer was found to be able to perform nontrivial algorithmic tasks. There are several notable differences between MixiT and the random transformer. Notably, MixiT has a principled random initialization, with provably stable forward pass, as shown in Theorem 2.1, whereas the random transformer uses standard random initialization.

Indeed, this leads to major differences in model behavior as the number of layer increases. Through in depth hyperparameter searches, we find that the random transformer does not scale well with respect to depth, confirming some of the original findings in Zhong and Andreas (2024). However, MixiT, with its carefully initialized random attention matrix designed to preserve signal propagation, does scale with respect to depth, suggesting that the random transformer suffers from signal propagation challenges and rank collapse without appropriate shaping (Dong et al., 2021; Noci et al., 2022; 2023).

| Model \ Depth | 2 | 8 | 16 |
|---|---|---|---|
| MixiT | 100% | 100% | 100% |
| Random Transformer | 100% | 23.53% | 22.88% |

Table 5: Performance comparison between MixiT and the random transformer, with respect to number of layers, on the decimal addition task. The random transformer's performance does not scale well with respect to depth, whereas attention matrix shaping helps MixiT scale with respect to depth.

These observations provide an explanation for the subspace selection hypothesis in Zhong and Andreas (2024), which refers to the phenomenon where random transformers operate in low-dimensional subspaces, i.e. a large fraction of the variance in hidden representations can be explained by the top principal components. We link this to the previously-studied phenomenon of representation collapse of transformer models at random initialization (Dong et al., 2021; Noci et al., 2022; 2023), where transformer hidden representations collapse to low-dimensional subspaces.

To substantiate this, we analyze the covariance between hidden representations for the language modeling task on Wikitext. This covariance is calculated between the last layer hidden representations within a sequence (higher covariance means the representations of different tokens are more similar), then averaged across sequences.

| Model \ # Layer | 2 | 4 | 8 | 16 | 32 |
|---|---|---|---|---|---|
| Random Transformer | 0.218 | 1.450 | 2.075 | 1.946 | 5.438 |
| MixiT | 0.088 | 0.260 | 0.210 | 0.154 | 0.104 |

Table 6: Covariance between hidden representations of different tokens for models of different depths, in unit $1e - 2$, on the Wikitext dataset. While the covariance for the random transformer steadily increases with respect to model size, it remains steady for MixiT.

Table 6 shows that this covariance for the random transformer steadily increases as the model size increases in depth, but remains steady for MixiT. This directly implies representation collapse in the random transformer, where the representation of different tokens becomes increasingly similar as the model size grows in depth.

This covariance degeneracy is especially catastrophic for language modeling, where deep models are required to achieve good performance. This helps explain why the Random Transformer struggles with language modeling and solving tasks when the model size increases beyond a certain number of layers (Zhong and Andreas, 2024).

**Implications for architecture design.** Notably, some of these results strengthen the argument for empirical approaches for architecture design found in previous work (Poli et al., 2024; Carstensen et al., 2025). In particular, Poli et al. (2024) uses performances in various synthetic tasks to design powerful hybrid architectures. Given out results on different model components having vastly different task-specific capabilities, can we design better mixture-of-expert models with *heterogeneous* components across experts (rather than the common symmetric architecture across experts)? And given the strong performance on Frozen-QK (and MixiT on some tasks), one can envision MoEs where some experts have trainable attention, but others having *random* attention. This can potentially perform well with a wide swath of tasks across the spectrum of reasoning-vs-memorization, with less overfitting and better generalization, while easing inference compute and KV caching for some tasks. Future work might also study hybrid training schedules, in which only a subset of architectural modules remain trainable – or are gradually unfrozen – which may strike an even better accuracy-efficiency trade-off.

## 6 RELATED WORK

**Static Attention.** Several studies explore simplified Transformers with frozen or random components. Notably, random Transformers with fixed layers but trainable embeddings can solve many algorithmic tasks (Zhong and Andreas, 2024). Similarly, replacing attention with fixed matrices - as in Synthesizer (Tay et al., 2021) or FNet (Lee-Thorp et al., 2021) retains competitive performance on certain benchmarks, suggesting learned attention is not always necessary. However, retrieval tasks often require flexible, input-dependent attention to form induction-like circuits. While their attention is also random, unlike our MixiT model it is input-*dependent* and so does not isolate the specific tasks for which attention is and is not needed. We relate our results to their work in detail in Section 5. Other past work has studied properties of other randomly frozen or lightly trained models, e.g. convolutional networks (Jarrett et al., 2009; Saxe et al., 2011; Arora et al., 2019), largely without focusing on specific tasks. In addition, (Hassid et al., 2022) also studies static attention, but uses the learned attention matrices over a reference corpus, and hence is not completely data-free, and does not have a stable signal propagation guarantee as MixiT. Indeed the completely data-free attention variants (Hassid et al., 2022) tested, without any reference corpus, perform poorly. Our work shows that such a variant's performance depends heavily on the task: on whether specialized circuits need to be formed.

**Stable Signal Propagation.** While there is a long line of work studying signal propagation in deep neural networks (Schoenholz et al., 2016; Poole et al., 2016; Lee et al., 2017; Yang and Schoenholz, 2017a), it was only more recently that Martens et al. (2021) introduced the concept that modifying activation functions can significantly improve stability of signal propagation, and leading to rapid training of large scale vision models without using normalization and skip connections (Zhang et al., 2022). This was later understood to yield a stable scaling limit, which is characterized by an SDE of the covariance matrix (Li et al., 2022). The covariance SDE framework is then used to understand how to design and shape general non-linearities like the Transformer self-attention (Noci et al., 2023), resolving the rank collapse issue (Dong et al., 2021; Noci et al., 2022) and has shown strong performance despite a much simplified Transformer block (He and Hofmann, 2023). Our theoretical result Theorem 2.1 also follows from this framework.

**Modular Tasks.** Several works have investigated the capabilities of Transformers in the classes of tasks analyzed here, including arithmetic (Nogueira et al., 2021). The role of feedforward layers in memorization in the Transformer architecture has been studied in Geva et al. (2020) and their inductive biases and scaling properties have been scrutinized (Bachmann et al., 2023). In this context, our work shows their relevance in conjunction with trainable or fixed attention. Orthogonal to our work, memorization has also been studied to understand generalization in neural networks (Zhang et al., 2016; Arpit et al., 2017; Anagnostidis et al., 2022).

**Mechanistic Interpretability.** Other closely related work is in the mechanistic interpretability, which aims to understand LLMs via examining and modifying their internals (Bereska and Gavves, 2024). Closely related is work related to identifying and understanding the behavior of induction heads (Olsson et al., 2022; Edelman et al., 2024; Bietti et al., 2023) and in-context learning (Chan et al., 2022). Our work demonstrates that the performance separation between input-dependent and input-*independent* attention is largely driven by the latter's inability to form induction heads. (Chen et al., 2024) also studies different components of the transformer model, but from a model dynamics point of view, analyzing differences in gradients between components, and on synthetic tasks such as indirect object identification and factual recall. Finally, Meng et al. (2022) which reverse-engineer how different Transformer components support behaviors like memorization, retrieval, and generalization.

**Efficient Attention.** A last area that has seen significant effort at understanding attention is that of efficient Transformers (Tay et al., 2022). While we study what happens when attention is replaced with a fixed input-independent matrix, this field has studied various useful aspects of the attention matrix such as attention sinks (Xiao et al., 2024) and compression (Kim et al., 2024). As our work demonstrates that for many tasks the full power of input-dependent attention is not needed, it may have its own implications for efficiency, e.g. by removing the need for the KV-cache.

## 7 CONCLUSION

In this work, we designed a spectrum of model architectures to systematically study the components of a transformer model. We found that, surprisingly, trainable attention is not required to form specialized circuits, and Frozen-QK can indeed perform well on language modeling. We also identified an obstruction towards scaling depth-wise in prior random models, and designed a principled remedy. Our work sheds important light on the functionalities of different model components, and shows that the transformer architecture has a built-in inductive bias towards in-context reasoning abilities.

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

APPENDIX

# A    PROOF OF THEOREM 2.1

**Theorem** (MixiT Covariance SDE (Exact))**.** *Consider the MixiT recursion $h_{\ell+1} = Attn(h_\ell)$ defined by (2.3) at initialization. Then as the width $n$ and depth $d$ go to infinity with $\frac{d}{n} \to \bar{\tau} > 0$, the upper triangular entries of the covariance matrix $\Phi_\ell = \frac{1}{n} h_\ell^\top h_\ell$ flattened to a vector in $\mathbb{R}^{m(m+1)/2}$ converges to the solution of the following SDE*

$$d\Phi_\tau = \left[ \frac{1}{m} \operatorname{Tr}(\Phi_\tau) - M(\Phi_\tau) \right] d\tau + [\Sigma^v(\Phi_\tau) + \Sigma^M(\Phi_\tau)]^{1/2} dB_\tau , \tag{A.1}$$

*where $M(\Phi) = \frac{1}{m^2} \sum_{\alpha\beta=1}^{m} \Phi^{\alpha\beta}$ is the average over all entries, $B_\tau$ is a standard Brownian motion in $\mathbb{R}^{m(m+1)/2}$, $\Sigma^v(\Phi)^{\alpha\beta,\gamma\delta} = \Phi^{\alpha\gamma}\Phi^{\beta\delta} + \Phi^{\alpha\delta}\Phi^{\beta\gamma}$ and*

$$\Sigma^M(\Phi)^{\alpha\beta,\gamma\delta} = \delta_{\alpha\gamma}C(\Phi^{\bullet\beta}, \Phi^{\bullet\delta}) + \delta_{\alpha\delta}C(\Phi^{\bullet\beta}, \Phi^{\bullet\gamma}) + \delta_{\gamma\beta}C(\Phi^{\bullet\delta}, \Phi^{\bullet\alpha}) + \delta_{\beta\delta}C(\Phi^{\bullet\alpha}, \Phi^{\bullet\gamma}) , \tag{A.2}$$

*where $\delta_{\alpha\gamma}$ is the Kronecker delta, $C(\Phi^{\bullet\beta}, \Phi^{\bullet\delta}) = \frac{1}{m}\langle \Phi^{\bullet\beta}, \Phi^{\bullet\delta}\rangle - \overline{\Phi}^{\bullet\beta}\overline{\Phi}^{\bullet\delta}$, $\Phi^{\bullet\beta} = [\Phi^{\alpha\beta}]_{\alpha=1}^m$ is the $\beta$-th column vector, and $\overline{\Phi}^{\bullet\beta} = \frac{1}{m}\sum_\alpha \Phi^{\alpha\beta}$ is the average.*

*Proof.* Firstly, we recall that based on Li et al. (2022), the linear network covariance matrix $\Phi_\ell = \frac{1}{n} h_\ell^\top h_\ell$ for the recursion $h_{\ell+1} = W_\ell^v h_\ell$ for $W_{\ell,ij}^v \sim \mathcal{N}(0, \frac{1}{n})$ satisfies the Markov chain

$$\Phi_{\ell+1} = \Phi_\ell + \frac{\Sigma^v(\Phi_\ell)^{1/2}\xi_\ell}{\sqrt{n}} , \tag{A.3}$$

where $\xi_\ell$ is a zero mean and identity covariance random variable, and the diffusion coefficient is $\Sigma^v(\Phi)^{\alpha\beta,\gamma\delta} = \Phi^{\alpha\gamma}\Phi^{\beta\delta} + \Phi^{\alpha\delta}\Phi^{\beta\gamma}$. Therefore, it is sufficient to isolate the contribution of the mixing component alone, and we will add the effect of the two components.

To this end, we consider the equivalent recursion

$$h_{\ell+1} = h_\ell \left( I_m + \frac{1}{\sqrt{nm}}(W_\ell^M - \bar{W}_\ell^M) \right) , \tag{A.4}$$

where we consider $W_{\ell,ij}^M \sim \mathcal{N}(0, 1)$ instead of $\mathcal{N}(0, \frac{1}{nm})$ due to the pre-factor, and $\bar{W}_{ij}^M = \frac{1}{m}\sum_{k=1}^m W_{kj}^M$ replaces each entry by its corresponding column average.

Next, we will observe that $\Phi_\ell$ satisfies a straight forward recursion

$$\begin{aligned}
\Phi_{\ell+1} &= \frac{1}{n} h_{\ell+1}^\top h_{\ell+1} \\
&= \left( I_m + \frac{1}{\sqrt{nm}}(W_\ell^M - \bar{W}_\ell^M) \right)^\top \Phi_\ell \left( I_m + \frac{1}{\sqrt{nm}}(W_\ell^M - \bar{W}_\ell^M) \right) \\
&= \Phi_\ell + \frac{1}{\sqrt{nm}} \left[ (W_\ell^M - \bar{W}_\ell^M)^\top \Phi_\ell + \Phi_\ell (W_\ell^M - \bar{W}_\ell^M) \right] \\
&\quad + \frac{1}{nm} (W_\ell^M - \bar{W}_\ell^M)^\top \Phi_\ell (W_\ell^M - \bar{W}_\ell^M) ,
\end{aligned} \tag{A.5}$$

which naturally separates itself into the diffusion and drift components via the coefficient scale of $\frac{1}{\sqrt{nm}}$ and $\frac{1}{nm}$ respectively.

We will compute the drift term next. Here we will drop some super and subscripts to reduce clutter, and write

$$\begin{aligned}
\sum_{\alpha,\beta=1}^{m} \mathbb{E}_\ell (W - \bar{W})^{\alpha\gamma} \Phi^{\alpha\beta} (W - \bar{W})^{\beta\delta} &= \sum_{\alpha\beta} \Phi^{\alpha\beta} \mathbb{E}_\ell \left[ W^{\alpha\gamma}W^{\beta\delta} - \frac{1}{m^2}\sum_{\mu\nu} W^{\mu\gamma}\bar{W}^{\nu\delta} \right] \\
&= \sum_{\alpha\beta} \Phi^{\alpha\beta} (\delta_{\alpha\beta}\delta_{\gamma\delta} - \frac{1}{m^2}\sum_{\mu\nu} \delta_{\mu\nu}\delta_{\gamma\delta}) \\
&= \delta_{\gamma\delta} \sum_{\alpha\beta} \Phi^{\alpha\beta} \left( \delta_{\alpha\beta} - \frac{1}{m} \right) ,
\end{aligned} \tag{A.6}$$

where $\mathbb{E}_\ell[\,\cdot\,] = \mathbb{E}[\,\cdot\,|\mathcal{F}_\ell]$ and $\mathcal{F}_\ell = \sigma(\{h_k\}_{k\le\ell})$, which translates to the final drift of

$$\frac{1}{n}\left(\frac{1}{m}\operatorname{Tr}(\Phi) - M_\Phi\right)I_n\,, \tag{A.7}$$

where $M_\Phi = \frac{1}{m^2}\sum_{\alpha\beta}\Phi^{\alpha\beta}$ is the average over all entries.

To calculate a single entry of the diffusion coefficient $\Sigma(\Phi)^{\alpha\beta,\gamma\delta}$, we will write $\widetilde{W} = W - \bar{W}$ and compute

$$\begin{aligned}
&\Sigma^M(\Phi)^{\alpha\beta,\gamma\delta}\\
&= \frac{1}{m}\sum_{\mu,\nu=1}^m \mathbb{E}_\ell(\widetilde{W}^{\mu\alpha}\Phi^{\mu\beta} + \Phi^{\alpha\mu}\widetilde{W}^{\mu\beta})(\widetilde{W}^{\nu\gamma}\Phi^{\nu\delta} + \Phi^{\gamma\nu}\widetilde{W}^{\nu\delta})\\
&= \frac{1}{m}\sum_{\mu,\nu}\mathbb{E}_\ell\left[\widetilde{W}^{\mu\alpha}\Phi^{\mu\beta}\widetilde{W}^{\nu\gamma}\Phi^{\nu\delta} + \widetilde{W}^{\mu\alpha}\Phi^{\mu\beta}\Phi^{\gamma\nu}\widetilde{W}^{\nu\delta} + \Phi^{\alpha\mu}\widetilde{W}^{\mu\beta}\widetilde{W}^{\nu\gamma}\Phi^{\nu\delta} + \Phi^{\alpha\mu}\widetilde{W}^{\mu\beta}\Phi^{\gamma\nu}\widetilde{W}^{\nu\delta}\right]\,.
\end{aligned} \tag{A.8}$$

At this point we focus on one term and compute

$$\begin{aligned}
\mathbb{E}_\ell\,\widetilde{W}^{\mu\alpha}\widetilde{W}^{\nu\beta} &= \mathbb{E}_\ell\,(W^{\mu\alpha} - \bar{W}^{\mu\alpha})(W^{\nu\beta} - \bar{W}^{\nu\beta})\\
&= \mathbb{E}_\ell\,(W^{\mu\alpha}W^{\nu\beta} - W^{\mu\alpha}\bar{W}^{\nu\beta} - \bar{W}^{\mu\alpha}W^{\nu\beta} + \bar{W}^{\mu\alpha}\bar{W}^{\nu\beta})\,,
\end{aligned} \tag{A.9}$$

$\delta_{\alpha\gamma}$ is the Kronecker delta, and we can separate further then compute

$$\begin{aligned}
\mathbb{E}_\ell\,W^{\mu\alpha}W^{\nu\beta} &= \delta_{\mu\nu}\delta_{\alpha\beta}\,,\\
\mathbb{E}_\ell\,W^{\mu\alpha}\frac{1}{m}\sum_{\nu'=1}^m W^{\nu'\beta} &= \frac{1}{m}\sum_{\nu'}\delta_{\mu\nu'}\delta_{\alpha\beta} = \frac{1}{m}\delta_{\alpha\beta}\,,\\
\mathbb{E}_\ell\,\bar{W}^{\mu\alpha}W^{\nu\beta} &= \frac{1}{m}\delta_{\alpha\beta}\,,\\
\mathbb{E}_\ell\,\bar{W}^{\mu\alpha}\bar{W}^{\nu\beta} &= \frac{1}{m^2}\sum_{\mu',\nu'=1}^m \delta_{\mu'\nu'}\delta_{\alpha\beta} = \frac{1}{m}\delta_{\alpha\beta}\,.
\end{aligned} \tag{A.10}$$

This implies

$$\mathbb{E}_\ell\,\widetilde{W}^{\mu\alpha}\widetilde{W}^{\nu\beta} = \delta_{\mu\nu}\delta_{\alpha\beta} - \frac{1}{m}\delta_{\alpha\beta} = (\delta_{\mu\nu} - \tfrac{1}{m})\delta_{\alpha\beta}\,. \tag{A.11}$$

At this point, we return to calculating $\Sigma^M(\Phi)^{\alpha\beta,\gamma\delta}$ and write

$$\begin{aligned}
\Sigma^M(\Phi)^{\alpha\beta,\gamma\delta} &= \frac{1}{m}\sum_{\mu\nu}(\delta_{\mu\nu} - \tfrac{1}{m})\delta_{\alpha\gamma}\Phi^{\mu\beta}\Phi^{\nu\delta} + (\delta_{\mu\nu} - \tfrac{1}{m})\delta_{\alpha\delta}\Phi^{\beta\mu}\Phi^{\gamma\nu}\\
&\quad + (\delta_{\mu\nu} - \tfrac{1}{m})\delta_{\gamma\beta}\Phi^{\mu\delta}\Phi^{\alpha\nu} + (\delta_{\mu\nu} - \tfrac{1}{m})\delta_{\beta\delta}\Phi^{\alpha\mu}\Phi^{\gamma\nu}\\
&= \delta_{\alpha\gamma}C(\Phi^{\bullet\beta}, \Phi^{\bullet\delta}) + \delta_{\alpha\delta}C(\Phi^{\bullet\beta}, \Phi^{\bullet\gamma}) + \delta_{\gamma\beta}C(\Phi^{\bullet\delta}, \Phi^{\bullet\alpha}) + \delta_{\beta\delta}C(\Phi^{\bullet\alpha}, \Phi^{\bullet\gamma})\,,
\end{aligned} \tag{A.12}$$

where $C(\Phi^{\bullet\beta}, \Phi^{\bullet\delta}) = \frac{1}{m}\langle\Phi^{\bullet\beta}, \Phi^{\bullet\delta}\rangle - \overline{\Phi}^{\bullet\beta}\overline{\Phi}^{\bullet\delta}$, $\Phi^{\bullet\beta} = [\Phi^{\alpha\beta}]_{\alpha=1}^m$ is the $\beta$-th column vector, and $\overline{\Phi}^{\bullet\beta} = \frac{1}{m}\sum_\alpha \Phi^{\alpha\beta}$ is the average.

To complete the proof, we will invoke the Markov chain convergence to SDE results in the Skorohod topology, see for example Li et al. (2022, Proposition A.6), which gives us the desired result.

$\square$

## B    PROOF OF THEOREM 5.1

**Theorem** (Universal Approximation of Frozen-QK). *Every continuous causal function with compact support can be approximated arbitrarily well by one layer of multihead attention and MLP, where query and key weight matrices are frozen at random initialization.*

*Proof.* The proof will follow the universal approximation theory for random feature regression in Banach spaces by Neufeld and Schmocker (2023), where we will set up the relevant Banach spaces.

Let the space of input sequences of length $m$, where each element is a vector in $V \subset \mathbb{R}^n$, where $V$ is compact, be denoted by $U = V^m$. The Banach space of continuous functions that map an input sequence from a compact set $K \subset U$ to an output sequence in $U$ is denoted by $(C(K, U), || \cdot ||_\infty)$, i.e. equipped with the sup-norm. The subspace of continuous **causal** functions, which we can denote as $C_{\text{causal}}(K, U)$, consists of all functions $f \in C(K, U)$ that satisfy the following property: For any given position $i \in \{1, \ldots, m\}$ and any two input sequences $u = (u_1, \ldots, u_m)$ and $v = (v_1, \ldots, v_m)$ in the domain $K$:

$$\text{If } u_j = v_j \text{ for all } j \leq i, \text{ then } (f(u))_i = (f(v))_i$$

This condition ensures that the output at position $i$ only depends on the input up to position $i$. Since this subspace is a closed linear subspace of the Banach space $C(K, U)$, it is itself a Banach space with the same supremum norm.

We define a random feature model based on a single multi-head attention layer with causal masking. Let the input be a sequence of hidden states $h \in \mathbb{R}^{n \times m}$, where $m$ is the sequence length and $n$ is the embedding dimension. The layer has $N_h$ attention heads. Conceptually, each attention head $i \in \{1, \ldots, N_h\}$ generates a single, matrix-valued **random feature**. This feature is the product of the hidden space $h$ and an attention pattern $\mathcal{A}_i$, which is a function that maps the input sequence $h$ to an $m \times m$ matrix. The randomness for each feature $\mathcal{A}_i$ comes from a pair of **frozen weight matrices**, $(W_i^Q, W_i^K)$, where $W_i^Q, W_i^K \in \mathbb{R}^{n \times d_k}$. These are drawn independently for each head from a random distribution (e.g. $\mathcal{N}(0, 1)$) at initialization and are not trained.

The random feature (attention pattern) for head $i$ is defined as:

$$g_i = h \, \mathcal{A}_i(h; W_i^Q, W_i^K) = h \, \text{Softmax}\left( \frac{(W_i^Q h)^\top (W_i^K h)}{\sqrt{d_k}} + M_{\text{causal}} \right), \tag{B.1}$$

where $M_{\text{causal}}$ is the causal mask matrix that prevents attention to future positions.

By Theorem 3 in Yun et al. (2020), Transformers are universal approximators for compactly supported sequence-to-sequence functions. Furthermore, as Neufeld and Schmocker (2023) shows that universal approximation can be lifted from deterministic feature functions to random feature functions in Banach spaces (Neufeld and Schmocker, 2023, Theorem 3.2), we can lift this universal approximation to the random features $g_i$, to approximate any causal function $f \in C_{\text{causal}}(K, U)$ to arbitrary precision.

Note that, given that both the value weight matrix $W_i^V$ of each head as well as the weights of the MLP layer are trainable, this is strictly more expressive than the random feature model we just constructed with $g_i$. Therefore one layer of self-attention and MLP must also be a universal approximator of causal functions, as desired.

$\square$

## C ADDITIONAL EXPERIMENTAL DETAILS

### C.1 HYPERPARAMETER SELECTION

We select for the optimal hyperparameters, shown in Table 8, for each task using a grid search. The search ranges for each task are shown in Table 7, determined a priori depending on task complexity, e.g. language modeling is inherently more complex than memorization.

| Task \ Hyperparameter | # Layers | Hidden dimension | # Heads | Learning rate | Batch size |
|---|---|---|---|---|---|
| Algorithmic | [2, 4, 8] | [512, 1024] | [4, 8, 16, 64] | [1e-3, 5e-4, 1e-4] | [128, 256, 512] |
| Retrieval | [2, 4, 8] | [512, 1024] | [4, 8, 16, 64] | [1e-3, 5e-4, 1e-4] | [256, 512, 1024] |
| $k$-hop | [5] | [256, 512, 1024] | [8] | [1e-4, 5e-4] | [128, 256] |
| Memorization | [2, 4, 8] | [512, 1024] | [4, 8, 16, 64] | [1e-3, 5e-4, 1e-4] | [256, 512, 1024] |
| Yelp | [4, 8] | [512, 1024] | [4, 8, 16, 64] | [1e-3, 5e-4, 1e-4] | [256, 512, 1024] |
| Language modeling | [8, 12] | [512, 1024] | [4, 8, 16] | [1e-3, 5e-4, 1e-4] | [256, 512, 1024] |

Table 7: Hyperparameter ranges used during grid search, for all architectures. Algorithmic tasks include decimal addition, Dyck-1 parentheses balancing, and modular addition.

Furthermore, we use a sequence length of 256 for Yelp sentiment classification and language modeling on Wikitext, and a sequence length of 2048 for Fineweb. Language modeling tasks are trained for 40,000 steps.

For the $k$-hop induction heads task, following (Sanford et al., 2024), we generate data using a sequence length of 100, and a maximum of $k = 16$ hops. We use a training set of 100,000 samples, a test set of 100 samples, and train for 5,000 steps.

For a fair comparison across models on the memorization task (Table 4), we use the same hidden dimension, with $L = 2$, and $n_h = 4$ heads. We train for $10,000$ steps with a learning rate of 0.005. Similarly, for a fair comparison on the retrieval task, reported in Figure 2, we fix the embedding dimension at $n = 1024$, with 4 heads and 2 layers, which suffice to learn induction heads (Elhage et al., 2021; Edelman et al., 2024). We search for the optimal learning rate for each architecture. Note that these two sets of results are meant to capture model performance when the accuracies are not saturated at 100%, to give a meaningful comparison between models.

| Task \ Hyperparameter | # Layers | Hidden dimension | # Heads | Learning rate | Batch size |
|---|---|---|---|---|---|
| Decimal Addition | 8 | 512 | 64 | 1e-3 | 128 |
| Dyck-1 | 4 | 512 | 64 | 1e-3 | 512 |
| Modular addition | 2 | 512 | 32 | 1e-3 | 256 |
| Retrieval | 2 | 1024 | 4 | 1e-4 | 1024 |
| $k$-hop | 5 | 512 | 8 | 1e-4 | 128 |
| Memorization | 2 | 1024 | 4 | 1e-3 | 256 |
| Yelp | 4 | 1024 | 16 | 5e-4 | 256 |
| Wikitext | 12 | 512 | 8 | 5e-4 | 512 |
| Fineweb-edu | 12 | 512 | 8 | 5e-4 | 512 |

Table 8: Optimal hyperparameters selected for MixiT.

# D    FURTHER RESULTS AND DISCUSSION

This section contains further discussion on experiments and results.

**Frozen-MLP Performance**    We conduct additional experiments on the performance of Frozen-MLP, with for tasks in Table 3 and Table 2. Note that originally, Table 3 and Table 2 were intended to showcase how varying the level of attention, from random, input-independent attention (MixiT), to random, input-dependent attention (Frozen-QK), to full standard attention, affects performance on these tasks. Hence Frozen-MLP was not included.

| Model \ Task | Wikitext $\downarrow$ | Fineweb-edu $\downarrow$ |
|---|---|---|
| Standard | 2.78 | 3.51 |
| Frozen-QK | 3.07 | 3.64 |
| MixiT | 3.73 | 4.27 |
| Frozen-MLP | 3.01 | 3.60 |

Table 9: Performance on language modeling tasks, in terms of log perplexity. Frozen-MLP performs competitively with the standard transformer. Note that as in Table 2, the sequence length used on Wikitext is 256, and the sequence length used on Fineweb is 2048. The results already reported in the main text are grayed out, but are included for comparison.

These are achieved with 12 layers, embedding dimension 512, and show that with sufficient model capacity, Frozen-MLP performs on par with the standard transformer. This is less surprising than it is for Frozen-QK, as Frozen-MLP clearly retains in-context reasoning capabilities given its trainable attention weights. Furthermore, the weights that are not frozen in Frozen-MLP, such as the embeddings layer and the values projectors, can likely ameliorate the lost knowledge storage capability from freezing MLPs, given sufficient hidden dimensions.

**Improved throughput for MixiT and Frozen-QK.**    MixiT and Frozen-QK exhibit a notable improvement in training time for language modeling. For instance, on the Fineweb-edu dataset, with the same architecture and hyperparameters on the same infrastructure, Frozen-QK trains 1267.1 samples per second on average, whereas the standard transformer

| Model \ Task | Decimal Addition↑ | Dyck-1↑ | Modular addition↑ | Memorization↑ | Yelp↑ |
|---|---|---|---|---|---|
| Standard | 98.58% | 95.80% | 100% | 100% | 90.55% |
| Frozen-QK | 100% | 97.38% | 100% | 100% | 90.86% |
| MixiT | 100% | 96.17% | 100% | 100% | 92.56% |
| Frozen-MLP | 99.91% | 95.92% | 100% | 98.1% | 90.63% |

Table 10: Frozen-MLP performance on algorithmic and sentiment classification tasks.

trains 1022.8 samples per second on average, while achieving similar log perplexities, 3.05 for standard and 3.15 for Frozen-QK. This represents a 23.9% improvement in training throughput, leading to a 23.9% speedup in terms of wall clock time. MixiT trains even faster, with 1349.0 training samples per second, or a 32.0% improvement in throughput. However, MixiT comes with noticeable degradation in perplexity.

**Performance with respect to number of heads for MixiT.** For some tasks, we observe that increasing the number of attention heads in MixiT can notably improve performance, as demonstrated by the decimal addition and Dyck-1 parentheses balancing tasks in Table 11. Intuitively, since each attention head uses a different random attention matrix, more attention heads gives the learnable MLP components more diverse attention patterns to choose from based on the input, hence lessening the disadvantage of static attention. Note that purely increasing the number of heads, without increasing the hidden dimension, reaches diminishing returns, as the per-head embedding dimension decreases proportionally, restricting expressiveness.

| Task \ Number of heads | 4 | 16 | 64 | 256 |
|---|---|---|---|---|
| Decimal addition | 34.71% | 34.70% | 50.72% | 91.87% |
| Dyck-1 | 77.93% | 81.68% | 89.38% | 91.83% |
| Yelp sentiment classification | 92.52% | 92.56% | 92.48% | 91.72% |

Table 11: Accuracy with respect to the number of heads on various tasks for MixiT. The hidden dimension is 512 for decimal addition and Dyck-1, and 1024 for Yelp sentiment classification. Increasing the number of heads increases the number of random attention matrices, giving learnable MLPs more diverse token mixing patterns to choose from based on the input, which can mitigate the disadvantage of static attention. However, a positive performance correlation does not appear in all tasks, such as in Yelp.

However, we do not observe this performance boost consistently across tasks. For instance, for Yelp sentiment classification, the performance is invariant with respect to the number of heads. This phenomenon remains interesting work for future study.

