# OpenReview forum: "Is Random Attention Sufficient for Sequence Modeling?"
_ICLR.cc/2026/Conference — Submitted to ICLR 2026_

### Official Review · Reviewer_Mct9 · 2025-10-15

**Soundness:** 3
**Presentation:** 2
**Contribution:** 3
**Rating:** 6
**Confidence:** 3

**Summary:**

This article investigates the model performance under fixed random QK, random attention score, and random MLP, and proves that some tasks do not require learnable QK, and some tasks do not even require learnable attention score.

**Strengths:**

1. The result is very interesting. The author conducted sufficient experiments to verify the impact of different degrees of weakening on sequence modeling. Even tasks such as induction heads can still be learned well when fixing Wq and Wk.

2. The author proposed a randomized attention score method and theoretically demonstrated its stability. In addition, the author also proved the universal approximation under random matrices.

**Weaknesses:**

1.
The presentation of MixiT is confusing.

From the experimental results, it can be seen that the performance of MixiT is not very good, especially in tasks related to language modeling. And only frozen-QK is well. So the "random attention" in your title is refers to frozen-QK only? I think if we could more clearly distinguish between random attention weight (frozen-QK) and random attention matrix (MixiT) in this article, it would be more conducive to reading.  Besides, the appearance of Theorem 2.1 in the seciton is also very abrupt, It should perhaps be placed in the section 5 where it will be used.

**Questions:**

1.
The boundary between what tasks Mixit can and cannot do well is very empirical. Do you have any further elaboration on this？On the other hand, I find that Mixit seems to be not able to do very well on long context tasks, such as induction heads and language modeling, Is that so?

2.
The results of Frozen-mlp are missing in Table 2 and Table 3.

---

> ### Author Response · Authors · 2025-11-25
> **Thank you for your review**
>
> Thank you for your helpful review. We very much hope we addressed all your concerns with the clarifications and new experiments. We are incorporating your suggestions into the paper.
>
> **Weaknesses**:
> > W1. The presentation of MixiT is confusing. Better differentiation between Frozen-QK and MixiT. Theorem 2.1 appears abruptly.
>
> **A**: The "random attention" in the title is deliberately left to be open-ended. Random attention can refer to either random input-independent attention (MixiT), or random input-dependent attention (Frozen-QK). Our work gives nuanced answers to the question: "Is random attention sufficient for sequence modeling?", in that input-independent random attention _is_ indeed sufficient for many sequence modeling tasks that can be solved without in-context reasoning, as evident in MixiT's strong performance in tasks such as those in Table 1. However, input-independent random attention is insufficient for tasks that require the formation of specialized circuits, whereas input-dependent random attention (Frozen-QK) indeed perform strongly on such tasks, which we show both experimentally (Tables 1, 2, and 3), and theoretically (Theorem 5.1). Thanks for raising this point. We will definitely point this out more explicitly in the updated paper.
>
> We see your point on Theorem 2.1 appearing to be abrupt. We are updating the lead-up to Theorem 2.1 to contain more context, including a more clear explanation of why the existence of the SDE solution implies stable signal propagation, to serve as a better segue between the MixiT model setup and the theorem about its properties.
>
> **Questions**:
> > Q1. Elaborate on MixiT capability beyond the empirical observations, and MixiT capability on long-context tasks.
>
> **A**:
> Indeed we provide ample empirical support for the separation between tasks that MixiT can and cannot do. Besides the empirical results, we use the proof of Theorem 5.1 to illustrate why MixiT cannot perform in-context reasoning: as the query and key vectors $Q_k$ and $K_k$ are input-independent in MixiT, the random feature $g_i = x \text{Softmax}(Q^T_k K_k)$ in MixiT is linear. Hence non-linear functions in $x$, such as the induction head function used for in-context reasoning, cannot be approximated. This is discussed in Section 5, and we will be sure to emphasize it further in the revised paper.
>
> Regarding your second point on long context tasks, it is indeed interesting to see if the performance conclusions for Frozen-QK and MixiT hold at greater sequence lengths. We conduct additional experiments using longer context lengths on language modeling, by using a greater level of both gradient accumulation and data parallelism. We test with sequence length 2048 on the Fineweb dataset, following the sequence length used in [1] on Fineweb. The log perplexities with 2k sequence length are:
>
> | Model | Frozen-QK  | MixiT | Standard |
> |-------|----------|-------------|-------------|
> | Fineweb ↓ | 3.64 | 4.27 | 3.51 |
>
> These results show that Frozen-QK's strong performance persists to the 2k sequence length, comparable to the standard transformer. MixiT, on the other hand, still struggles. These results are consistent with our findings at shorter context lengths.
>
> Note that induction heads and language modeling tasks can have either long or short sequence length. The main differentiating factor in MixiT's performance is whether in-context learning is required. The fact that MixiT does not perform well when in-context reasoning is required, at either long or short context length, is itself useful, allowing MixiT to serve as a litmus test for when in-context reasoning is necessary for a task.

---

> > ### Author Response · Authors · 2025-11-25
> > **[Response continued]**
> >
> > > Q2. The results of Frozen-mlp are missing in Table 2 and Table 3.
> >
> > **A**: Thank you for this point. We conducted additional experiments on Frozen-MLP, the results on language modeling:
> > | Model | Wikitext ↓ | Fineweb-edu ↓ |
> > |-------|----------|-------------|
> > | Frozen-MLP | 3.01 | 3.14 |
> >
> > These are achieved with 12 layers, embedding dimension 512, and show that with sufficient model capacity, Frozen-MLP performs on par with the standard transformer. This is less surprising than it is for Frozen-QK, as Frozen-MLP clearly retains in-context reasoning capabilities given its trainable attention weights. Furthermore, the weights that are not frozen in Frozen-MLP, such as the embeddings layer and the values projectors, can likely ameliorate the lost knowledge storage capability from freezing MLPs, given sufficient hidden dimensions.
> >
> > For the tasks in Table 3, we have:
> >
> > | Task | Decimal Addition  | Dyck-1 | Modular addition | Memorization |Yelp |
> > |-------|----------|-------------|-------------|-------------|-------------|
> > | Frozen-MLP | 99.91 | 95.92 | 100 | 98.1 | 90.63 |
> >
> > These results show that Frozen-MLP performs on par with the standard transformer, except in the pure memorization task (when the model specification falls within the hyperparameter search ranges).
> >
> > (Note that originally, Table 2 and Table 3 were intended to showcase how varying the level of attention, from random, input-independent attention (MixiT), to random, input-dependent attention (Frozen-QK), to full standard attention, affects performance on these tasks. Hence Frozen-MLP was not included. But we will include these new results in the updated paper.)
> >
> > Thank you again for your constructive review, which helped us improve our paper. We'd be more than happy to continue the discussion, if you have any additional comments or questions.
> >
> > [1] Penedo et al. The FineWeb Datasets: Decanting the Web for the Finest Text Data at Scale.

---

> ### Author Response · Authors · 2025-12-03
> **Summary for AC**
>
> Dear AC:
>
> Here we summarize the main changes to the paper in response to reviewer comments:
>
> - Ran additional language modeling experiments at longer sequence lengths. The competitive performance of Frozen-QK (compared with the standard transformer) still holds, indicating its robustness.
> - Conducted additional experiments on sensitivity of MixiT to different random initializations of its attention matrix, confirming MixiT performed worse with the standard initialization, compared with the one derived in Theorem 2.1.
> - Clarified MixiT as a resolution of the subspace selection phenomenon in Section 5.
> - Added to MixiT's usefulness beyond this study, e.g. for architecture design: can potentially design MoEs with heterogeneous components across experts, some with even _random_ attention, to improve both efficiency and generalization.
> - Updated the lead-up to Theorem 2.1 to contain more context, including a more clear explanation of why the existence of the SDE solution implies stable signal propagation, to serve as a better segue between the MixiT model setup and the theorem about its properties.
> - Conducted additional experiments on Frozen-MLP.
>
> More minor updates:
> - Clarified contribution of MixiT as an improved, more principled random transformer, and as a tool for understanding the standard transformer.
> - Clarified contribution around the role of MLPs.
> - Abbreviated Theorem 2.1 in the main text, to improve readability.
>
> We believe these changes addressed all reviewer concerns. If you have any additional questions or comments, we'd love to continue the discussion! Thank you!

---

### Official Review · Reviewer_Cdgz · 2025-10-30

**Soundness:** 3
**Presentation:** 3
**Contribution:** 2
**Rating:** 4
**Confidence:** 4

**Summary:**

The paper
- finds that Frozen-QK with random attention weights can perform competitively with the standard transformer on language modeling tasks. It's expressiveness is also enough for a wide class of sequence-level functions.
- proposes MixiT, proves its training stability. It achieves performance comparable to fully trained tf and Frozen-QK expect induction heads tasks and language modeling.

**Strengths:**

The paper did abundant experiments and derived theory to support the papers claim, which makes the paper sound.

**Weaknesses:**

W1: Your main claim is that the performance separation between input-dependent and input-independent attention is largely driven by the latter's inability to form induction heads. However, there are some papers [1], [2] explaining the mechanistic of induction heads theoretically which I think is missing from your paper. In their papers, they all have data-dependent attention for the second layer and it must contribute to the proof. What is the contribution of your paper to the (theoretical) understanding of the mechanics?

W2: You proved Frozen-QK has enough expressivity, but it does not answer the question how can Frozen-QK form the induction head. The former doesn't guarantee the latter because the latter question is about optimization.

[1] Nichani, E., Damian, A., & Lee, J. D. (2024). How transformers learn causal structure with gradient descent. arXiv preprint arXiv:2402.14735.

[2] Chen, S., Sheen, H., Wang, T., & Yang, Z. (2024). Unveiling induction heads: Provable training dynamics and feature learning in transformers. Advances in Neural Information Processing Systems, 37, 66479-66567.

**Questions:**

Q1: I didn't understand your third contribution and didn't find the part of the paper it correlated to.

Also see the weaknesses above.

---

> ### Author Response · Authors · 2025-11-25
> **Thank you for your review**
>
> Thank you for your review. We very much hope we addressed all your concerns with the clarifications. We are incorporating your suggestions into the paper.
>
> **Weaknesses**:
> > W1: Does not explain the mechanistics of forming induction heads.
>
> **A**: Thank you for the question. You are correct that we do not provide a theoretical analysis of the mechanism of which self-attention heads learn important features, such as forming an induction head. However, that is also not our goal, as we simply wanted to demonstrate that (1) random-attention-weight heads can still perform competitively on sequence modeling tasks, which was unexpected, and (2) provide a possible reason that explains the surprising performance. Our theoretical contribution towards this direction is that we demonstrated that while not necessarily efficient, it remains possible to approximate an induction head (function) using a large number of Frozen-QK heads.
>
> Note that beyond the performance separation between input-dependent and input-independent attention that we show theoretically and empirically, we also contribute a principled understanding of signal propagation in random transformer models.
>
> > W2: You proved Frozen-QK has enough expressivity, but it does not answer the question how can Frozen-QK form the induction head. The former doesn't guarantee the latter because the latter question is about optimization.
>
> **A**: This is a great point that distinguishes the difference between expressivity and trainability, which is true in general. However, we would like to highlight that our proof treats the Frozen-QK heads as random features. The intermediate architecture that we introduced in our proof, which simply performs linear (random feature) regression based on Frozen-QK heads, can be shown to form an approximate induction head since the optimization problem is convex in the limit. The only additional caveat is that we are layering on top of Frozen-QK heads with an MLP layer instead, and the only job of the MLP layer is to learn linear regression coefficients. As this is easily achievable in practice, it indeed gives reason to believe that an approximate induction head is learnable given enough Frozen-QK heads via standard first order methods. And indeed, our experiments show that Frozen-QK can learn induction heads in practice (Tables 1, 2, 3; Figure 3).
>
>
> **Questions**:
> > Q1: I didn't understand your third contribution and didn't find the part of the paper it correlated to.
>
> **A**: The third contribution refers to providing an explanation to the "subspace selection phenomenon" in [1], where [1] found that random transformers operate in low-dimensional subspaces, i.e. a large fraction of the variance in hidden representations can be explained by the top principal components. We link this to the previously-studied phenomenon of representation collapse of transformer models at initialization [2, 3, 4], where transformer hidden representations collapse to a proper subspace.
>
> This is supported by both the subspace selection observations in [1], as well as our analyses of the covariance of hidden representations (Table 6), where the covariance of hidden representations in prior random transformers grow with respect to model depth (signaling the representations are becoming more homogeneous).
>
> Our proposed solution, MixiT, is a random transformer with a provably stable forward pass, i.e. the hidden representations do not collapse as the model size increases in depth. Correspondingly, the covariance of hidden representations remains stable for MixiT (Table 6), and performance does not degenerate as the model grows deeper (Table 5), unlike prior random transformers. ([1] in fact observed that prior random transformers with 2 layers outperform those with 4 layers.) This discussion is elaborated in the "Relation to random transformers" subsection (under Section 5).
>
> Thank you for raising this point, we are updating the paper to clarify this.
>
> Thank you again for your review, we'd love to continue the discussion, if you have any additional comments or questions.
>
> [1] Zhong and Andreas. Algorithmic Capabilities of Random Transformers.
>
> [2] Dong et al, Attention is Not All You Need, Pure Attention Loses Rank Doubly Exponentially with Respect to Depth.
>
> [3] Noci et al, Signal Propagation in Transformers, Theoretical Perspectives and the Role of Rank Collapse.
>
> [4] Noci et al, The Shaped Transformer, Attention Models in the Infinite Depth-and-Width Limit.

---

### Official Review · Reviewer_5cHZ · 2025-10-31

**Soundness:** 3
**Presentation:** 3
**Contribution:** 3
**Rating:** 6
**Confidence:** 4

**Summary:**

The work studies what aspects of transformer performance rely on the self-attention mechanism by comparing the transformer to variants with frozen attention weights or MLP layers. They find that the transformer with frozen QK layers can achieve competitive performance on the induction head and language modeling tasks. The authors also propose the Mixing Transformer (MixiT) which uses random attention and overcomes prior depthwise scaling challenges of random transformers.

**Strengths:**

- The writing and methodology are clear and easy to follow.
- The experiments cover a diverse set of tasks that highlight which components are most important.
- The finding that the Frozen-QK model can form induction-head-like behavior and achieve strong performance on retrieval and k-hop tasks is particularly interesting.

**Weaknesses:**

- The discussion of MLPs being important for knowledge storage mostly confirms prior findings and does not extend existing insights.
- The motivation and practical usefulness of the MixiT architecture are not entirely clear. The paper emphasizes that the MixiT architecture provides stable signal propagation as compared to a random transformer, but still underperforms on the induction heads task and on language modeling.
- For the language modeling experiments the sequence length is 256 which is quite short. Do results hold for longer sequence length?

**Questions:**

- How sensitive are the Frozen-QK and MixiT models to the specific random initialization used?
- How do the authors see MixiT being used beyond this study -- as a diagnostic model or are there some implications for architecture design?

---

> ### Author Response · Authors · 2025-11-24
> **Updated results and clarifications**
>
> Thank you for your insightful review. We very much hope we addressed all your concerns with the clarifications and new experiments. We are incorporating your suggestions into the paper.
>
> **Weaknesses**:
> > W1: Discussion on MLPs not novel.
>
> **A**:
> Indeed, the fact that MLPs play a crucial role in knowledge storage is not new. Our work sheds more nuanced insight on the interplay between MLPs and attention, in that attention actually assists MLPs on knowledge storage, as evident in the fact that the bits-per-parameter (normalized for the number of parameters) stored knowledge drops from 2.98 to 2.25, from the standard transformer to Frozen-QK. This is a debate far from settled [1, 2, 3].
>
> > W2: Clarify motivation and practical usefulness of the MixiT architecture.
>
> **A**: We regard MixiT both as an improved random transformer and as a tool for understanding the standard transformer, _not_ as a state-of-the-art model architecture.
> Indeed, MixiT's more stable signal propagation is shown both theoretically (Theorem 2.1) and empirically (Sections 2 and 4), as evident in both MixiT's performance in Table 3, and the fact that it does not suffer from rank collapse (Table 5 and 6). This provides an answer to the subspace selection hypothesis in prior work [4], as well as prescribes how random transformers can be scaled up, without the representation collapse seen in previous random transformers [4].
>
> Note that MixiT underperforms Frozen-QK and the standard transformer on the induction heads task and on language modeling (it does not underperform previous random transformers). This in itself is a signal on the relevance of in-context reasoning for a given task.
>
> Furthermore, MixiT can serve as a litmus test for whether in-context reasoning is required for a given task, in that if MixiT performs well on a task, then a solution that does not require in-context reasoning exists and can be learned. These points are discussed in Sections 5 and 6, and we are updating the paper to further emphasize them.
>
> > W3: Do results hold for longer sequence length beyond 256?
>
> **A**: It is indeed interesting to see if the performance conclusions for Frozen-QK and MixiT hold at greater sequence lengths. We updated our code with a greater level of gradient accumulation and data parallelism, and tested on sequence length 2048 on the Fineweb dataset, following the sequence length used in [5]. The log perplexity on Fineweb with 2k sequence length:
>
> | Model | Frozen-QK  | MixiT | Standard |
> |-------|----------|-------------|-------------|
> | Fineweb ↓ | 3.64 | 4.27 | 3.51 |
>
> These results show that Frozen-QK's strong performance persists to the 2k sequence length. Note that this perplexity is higher than the perplexity with sequence length 256 (3.16 for Frozen-QK), which is analogous to previously observed performance degradation with respect to context length, e.g. in [6].
>
> **Questions**:
> > Q1: Frozen-QK and MixiT sensitivity to specific random initializations.
>
> **A**: We expect MixiT to perform worse with a different random initialization for the random attention matrix, as the MixiT initialization in Theorem 2.1 is specifically derived to ensure stable signal propagation when the attention matrix is entirely random.
> To empirically validate this, we set the MixiT random attention matrix with the standard PyTorch initialization, and test on Yelp sentiment classification and on the algorithmic task modular addition.
>
> | Model | Yelp Acc ↑ | Modular Addition ↑  |
> |-------|----------|-------------|
> | MixiT with **Standard** Init | 87.55 | 97.3 |
> | MixiT with Theorem 2.1 Init | 92.56 | 100 |
>
> These results indicate that the initialization in Theorem 2.1 indeed helps MixiT perform well. (Note that the theorem is proven in the limit, hence the initialization's benefits are expected to become more pronounced for deeper models.) These results are consistent with prior findings such as [7], which found that static random attention typically leads to poor results.
>
> Note our derived stable initialization in Theorem 2.1 pertains only to MixiT; we use the standard initialization for Frozen-QK. As a result, Frozen-QK's sensitivity to the initialization is at the same level as standard models. While this has been studied before [e.g. 8, 9] and is out of scope for our work, we sanity check Frozen-QK's sensitivity by using different initializations for all Linear modules, Xavier and Kaiming (Kaiming initialization is very similar to the default PyTorch Linear initialization, with a different gain factor):
> | Model | Yelp Acc ↑ | Modular Addition ↑ |
> |-------|----------|-------------|
> | FrozenQK with Xavier Init | 90.71  | 100 |
> | FrozenQK with Kaiming Init | 90.56 | 100 |
> | Originally reported | 90.86 | 100 |
>
> These results indicate that Frozen-QK's performance is much less sensitive to the initialization. Thank you for this question. We will include these new results in the updated paper.

---

> > ### Author Response · Authors · 2025-11-24
> > **[Response continued]**
> >
> > > Q2: MixiT usefulness beyond this study.
> >
> > **A**: As a diagnostic tool, MixiT can serve as a litmus test for when in-context learning is necessary for learning a solution, as it *is* able to to scale depth-wise given its stable signal propagation. And our work has exciting implications for architecture design and training. For instance, are there better training paradigms, such as during task-dependent post-training, when we can *gradually* unfreeze different model components, depending on where the task falls under the in-context-learning-vs-memorization spectrum. As our work confirms that not all parameters are created equal for a given task, hence it follows that they don't require the same level of training. This can lead to better generalization by mitigating overfitting of parameters that need less training.
> >
> > Furthermore, can we design better MoEs with *heterogeneous components across experts* (rather than the common symmetric architecture across experts), given that different model components have such different capabilities? And given the strong performance on Frozen-QK (and MixiT on some tasks), one can envision MoEs where some experts have trainable attention, but others having random attention. This can potentially perform well with a wide swath of tasks across the spectrum of reasoning-vs-memorization, with less overfitting and better generalization, while easing inference compute and KV caching for some tasks.
> >
> > Thank you for raising this point, we will update our discussion around these points.
> >
> > [1] Chen et al. Knowledge localization: Mission not accomplished? Enter query localization!
> >
> > [2] Yao et al. Knowledge circuits in pretrained transformers.
> >
> > [3] Yang et al. Knowledge Neurons: Editing Large Language Models via Knowledge Neurons.
> >
> > [4] Zhong and Andreas. Algorithmic Capabilities of Random Transformers.
> >
> > [5] Penedo et al. The FineWeb Datasets: Decanting the Web for the Finest Text Data at Scale.
> >
> > [6] Du et al. Context Length Alone Hurts LLM Performance Despite Perfect Retrieval.
> >
> > [7] Hassid et al. How much does attention actually attend? Questioning the importance of attention in pretrained transformers.
> >
> > [8] Kaplan et al. Scaling Laws for Neural Language Models.
> >
> > [9] Ainsworth et al. Git Re-Basin: Merging Models Modulo Permutation Symmetry.

---

### Meta-Review · Area_Chair_qGiC · 2026-01-06

**Summary:**

The paper studies the impact of random attention weights on sequence modeling. It freezes different (random) components of transformer, and studies how much it contributes to the performance gains.

**Reviewer Concerns:**

- The paper does not adequately connect to existing theoretical work on induction heads, nor clarify how its results advance mechanistic understanding beyond prior analyses.
- The purpose and practical usefulness of the MixiT architecture are not clearly articulated, and the boundary between tasks where MixiT succeeds or fails appears largely empirical, with no deeper explanation.
- While frozen-QK is shown to be expressive enough in principle, the paper does not explain how such architectures would form induction heads through training.

While the authors' responses address some of the concerns partially, the insights derived from these empirical results are marginal.

**Reviewer Scores:**

The original scores were 6/6/4. The reviewers would likely maintain their ratings.

---

### Decision · Program_Chairs · 2026-01-26

Reject